# Leukoplakia in the Oral Cavity and Oral Microbiota: A Comprehensive Review

**DOI:** 10.3390/cancers13174439

**Published:** 2021-09-03

**Authors:** Giacomo Pietrobon, Marta Tagliabue, Luigi Marco Stringa, Rita De Berardinis, Francesco Chu, Jacopo Zocchi, Elena Carlotto, Susanna Chiocca, Mohssen Ansarin

**Affiliations:** 1Division of Otolaryngology and Head and Neck Surgery, European Institute of Oncology IRCCS, 20142 Milan, Italy; giacomo.pietrobon@ieo.it (G.P.); rita.deberardinis@ieo.it (R.D.B.); francesco.chu@ieo.it (F.C.); jacopo.zocchi@ieo.it (J.Z.); mohssen.ansarin@ieo.it (M.A.); 2Department of Biomedical Sciences, University of Sassari, 07100 Sassari, Italy; 3Unit of Otolaryngology, University of Ferrara, 44121 Ferrara, Italy; strlmr@unife.it; 4Department of Otorhinolaryngology, Fondazione IRCCS Policlinico San Matteo, 27100 Pavia, Italy; elena.carlotto01@universitadipavia.it; 5Department of Experimental Oncology, European Institute of Oncology IRCCS, 20139 Milan, Italy; susanna.chiocca@ieo.it

**Keywords:** oral cavity leukoplakia, microbiota, microbiome, oral cavity cancer, dysplasia, premalignant disorder

## Abstract

**Simple Summary:**

The aim of this narrative review is to better understand the role of the oral microbiota in oral cavity leukoplakia. We provide a comprehensive review, exhaustively summarizing the steps taken in this field.

**Abstract:**

We reviewed the current published literature on the impact of oral microbiota on oral cavity leukoplakia (OLK), aiming at clarifying its role in disease transformation. The analysis unveiled that bacterial richness and diversity in the oral cavity tend to be decreased in OLK compared to healthy controls, with a reduction in the prevalent commensals, such as *Streptococci*, and elevation of anaerobes. Moreover, *Fusobacterium nucleatum*, *Porphyromonas gingivalis* and *Prevotella intermedia* are recurrent findings, and they already have been linked to periodontal disease. These microbial community changes may also represent a marker for the transition from OLK to oral squamous cell carcinoma. Unfortunately, the reviewed studies present several limitations, making an objective comparison difficult. To overcome these biases, longitudinal studies are necessary.

## 1. Introduction

The oral cavity is a complex biologic environment, where countless microorganisms interact continuously with one another and the mucosal epithelium, maintaining a delicate balance [1,2]. More than 700 different bacterial species have been identified in the oral cavity, and likely more will be discovered in the following years, thanks to the ever-evolving technologies (e.g., Next Generation Sequencing (NGS)). Fungi, archaea and viruses complete the oral flora, but the knowledge about their role is still limited. The microbiome is usually but erroneously identified with the bacteriome. The term “dysbiosis” refers to the perturbation of the microbial ecosystem, which is modified constitutionally and functionally. This perturbation could be associated with oral disease [2]. The meaning of this association has become the focus of plenty of studies in the last decade, especially as far as oral cancer is concerned. Tobacco, alcohol and areca nut (betel) are known risk factors for oral squamous cell carcinoma (OSCC), but a percentage of patients lack exposition. Oral dysbiosis associated with periodontal disease has gained attention as a possible etiologic factor for OSCC, either isolated or synergistic with other agents.

Chronic inflammation induced by microorganisms is implicated in cancer of other sites (e.g., *Helicobacter pylori* in the stomach). Thus, the mechanism may replicate in the oral cavity as well. Several studies proved a different bacterial enrichment of samples from patients affected by OSCC. Specifically, two bacteria, *Fusobacterium nucleatum* and *Porphyromonas gingivalis*, support carcinogenesis in a murine model [3]. Despite all the reports, a causal relationship between oral dysbiosis and oral cancer has not been demonstrated yet. However, two theories have been proposed: in the former, the so-called “bacteria before tumour”, bacterial damage to the epithelial cells activate a cascade of inflammatory pathways, leading to cell replication and reactive oxygen species (ROS) production and, ultimately, to DNA damage and carcinogenesis.

In the latter, the so-called “bacteria after tumour”, opportunistic bacteria are attracted by the hypoxic, hyper vascularized tumour environment, and they sustain the progression of the unhealthy ecosystem [1].

Oral cavity leukoplakia (OLK) is defined as a “predominantly white plaque of questionable risk, having excluded (other) known diseases or disorders that carry no increased risk for cancer” [4] (Figure 1). It is a premalignant disorder with a reported risk of cancerization ranging from 1% to more than 30% [5]. To date, the grade of dysplasia (SIN 1-2-3) defined by the pathologist on the biopsy specimen remains arguably the best estimator of malignant transformation [6]. However, aged non-smoker females are characterized by an increased risk [7].

Because oral cancer is the final event of a progressive multistep process, research has focused on the relationship between oral premalignant disorders (OPMDs) and the microbiome, aiming to assess any possible etiologic role of the latter in identifying any new marker of prognosis. Study outcomes have been conflicting, with some reporting similarities between OPMD and OSCC and others showing variations instead [1,8,9]. To note, a comparison between these studies is hampered by the different sampling of the oral cavity (biopsy vs. swab vs. mouth wash).

This narrative review will evaluate the current available evidence on the relationship between the oral microbiome and oral leukoplakia. We also clarify some critical aspects of this debated topic.

We originally planned a systematic review of the studies investigating the relationship between oral microbiota and OLK published up to May 2021. The search was conducted in the electronic databases of Pubmed, Scopus and Embase, using the following keywords: “oral leukoplakia” or “oral cavity leukoplakia” OR “oral premalignant disorder” AND “microbiome” OR “micriobiota”. However, only 13 consistent papers were retrieved, and all of them were highly inhomogeneous. For this reason, we opted for converting the study into a narrative review, based on these 13 papers, but further expanded to any related works, to possibly provide the most accurate picture of the current scientific knowledge about the association between oral premalignant disorders, mainly leukoplakia, and the oral microbiota.

## 2. Discussion

### 2.1. The First Dilemma: What to Sample

Every portion of the oral cavity is inhabited by plenty of microorganisms, with differences related to the structure (keratinized vs. non-keratinized) and state of the mucosa [10]. DNA may be extracted by either tissue biopsies [11] or unstimulated saliva, called whole mouth fluid (WMF) [12], or mouth rinse or mucosal swabs. The latter may be taken solely from the OLK or, conversely, from all different oral subsites [13,14]. Controls, if present, are represented by unaffected subjects, in case saliva is used, or oral contralateral healthy mucosa, when possible.

Results of such variable samplings provide different information, which is difficult to compare among studies. Saliva and multiple-site swabs picture the microbiome of the whole mouth, while localized swabs or tissue samples focus on the single oral lesion. Amer et al. reported significant changes in the abundance of three of the six most common oral phyla in OLK compared to healthy mucosa from the same patients [7]. Similarly, Decsi et al. found 43 different species in OPMD compared to 18 species in the healthy mucosa, with only 8 shared by both tissues [11]. These results question the validity of whole-mouth sampling in oral leukoplakia since it may be altered by periodontal disease or dental caries [12].

Nonetheless, other saliva-based studies were able to identify significant differences in the microbiome composition between OLK and healthy controls [12,15,16,17,18]. Moreover, although through a highly biased study (i.e., time lag between OLK and sampling; lack of histologic assessment of OLK), Shridhar et al. showed a better proportion of quantifiable pathogens from salivary rinse rather than gingival swabs for three selected species [18]. The authors then advocate the use of salivary rinse for population-based studies in low-resource settings.

So far, we have lacked conclusive evidence about the most suitable oral cavity sampling for microbiome assessment. Answers will likely come from a prospective comprehensive study, including and comparing simultaneously each of the methods available.

### 2.2. The Foreground: Bacteria

To date, bacteria represent the vast majority of the microbiome under investigation, more than viruses or fungi, with more than 700 species and 13 phyla identified. Through NGS, it is now possible to amplify 2 to 3 of the variable regions (V1 to V5) of the 16s rRNA gene, a bacterial fingerprint. Each operating taxonomic unit (OTU) obtained from the process is compared with the Human Oral Microbiome Database (HOMD) [19]. A sequence identity of at least 98% is necessary to identify the bacterial species [7].

Table 1 summarizes the most relevant findings of the original studies available on oral leukoplakia and the microbiome up to May 2021. Papers retrieved in the original search but dealing exclusively with OSF were excluded from the table. To note, part of these reports also included patients affected by OSCC or exposed to renowned risk factors (mainly alcohol and areca/betel nut). Results are highly variable, and a “fil rouge” is missing.

As far as biodiversity is concerned, only a few studies have investigated alfa-diversity, and they detected either a high bacterial richness [7], depletion [11,15] or even equivalence [12,16] between the OLKs and healthy controls. Instead, beta-diversity has been commonly reported between the same two groups [7,8,12,15], although in some cases the results were difficult to interpret because of different subsite [14] or subgroup analyses [13].

Bacteria related to OLKs are different across studies, and specification is more often limited to phyla and genera than species. Ganly et al. found an enrichment of the periodontal pathogens *Fusobacterium*, *Prevotella* and *Alloprevotella* but depletion of commensal bacteria (*Streptococcus*) along the sequence “controls → OLK → OSCC”. They also reported *Fusobacterium* and *Veilonella* to be more abundant in OLK compared to healthy controls [16]. Amer et al. discovered reduced levels of *Firmicutes* and increased abundance of *Fusobacterium nucleatum*, *Leptotrichia* spp., *Campylobacter* spp. and *Rotha mucilaginosa* in OLK relative to contralateral healthy mucosa. Conversely, the latter showed enrichment in *Streptococcus* species (particularly *S. Mitis*) and *Gemella haemolysans* [7]. Similar results were reported by Decsi et al., which is increased *Fusobacterium nucleatum* and decreased *Streptococcus mitis* in OLKs compared to healthy mucosa [11]. In the study by Hu et al., OLKs were associated with reduced *Firmicutes* and increased *Bacteroidetes* and *TM7* (also known as *Saccharibacteria*), and the prevalent genus was *Haemophilus* [15]. Similarly, Gopinath showed increased *Bacteroidetes* and reduced *Firmicutes* in OLK, which were further discriminated by the tetrad of *Salmonella*, unclassified *Enterobacteriaceae*, *Prevotella* and, above all, *Megasphera*. They also found reduced levels of *Granulicatella* and *Porphyromonas gingivalis* in OLK compared to OSCC [12]. Hashimoto et al. observed an increased abundance of *Porphyromonas gingivalis* and *Streptococcus anginosus* in OLK and OSCC relative to the controls [17]. Based on previous reports, Shridhar et al. did not investigate the whole microbiome in their population-based study. Instead, they focused on three renowned bacterial pathogens: *Porphyromonas gingivalis*, *Fusobacterium nucleatum* and *Prevotella intermedia* [18].

In the end, most studies consistently detected the association of OLKs with reduced levels of *Firmicutes*. They found increased abundance of anaerobic bacteria, mainly *Fusobacterium nucleatum*, *Prevotella intermedia* and *Porphyromonas gingivalis*, all of which are well established periodontal pathogens prone to thrive in a hypoxic environment. The phylum *Firmicutes* includes the genus *Streptococcus*, the main oral commensal, and whose depletion is often reported in regard to premalignant and malignant lesions [7,11,15,16,17]. Of note, the disease is unlikely correlated to one single bacterium, and we should instead consider clusters or communities of bacteria, both pathogens and non-pathogens [7,8,20]. As clearly pointed out by Ganly et al., these two bacterial categories display a tight interaction, which may be either collaborative with “friends” and inhibitive with “enemies” [16].

Nonetheless, it would be too simplistic to classify bacteria as pure pathogens. For example, both *Fusobacteria* and *Bacteroidetes* are usually low abundance oral commensals, and their pro-inflammatory response could be dose-dependent once a critical threshold is overpassed [21]. For this reason, it is premature to consider the elimination of bacteria such as *Fusobacterium nucleatum* to prevent the associated conditions [22].

### 2.3. The Background: Fungi, Viruses and Archaea

Most studies on oral leukoplakia and the microbiome neglect microorganisms other than bacteria, although over 100 different commensal fungal species already have been identified in the oral cavity, including *Candida*, *Aspergillus*, *Fusarium* and *Cryptococcus* [23].

Candida is by far the most studied fungus of the oral cavity, because it causes several different lesions, particularly pseudomembranous candidiasis (also known as “thrush”), which is related mainly to immunosuppression (e.g., HIV infection, diabetes mellitus, chemotherapy) or bacterial-depleting antibiotic treatments [5].

Amer et al. found significant colonization by *Candida* species in 35% of OLK samples, particularly from the tongue and palate; this percentage was higher than the contralateral healthy mucosa (20%) and healthy controls (13.5%). They also reported that 70% of patients in two out of five bacterial clusters were *Candida*-positive, while the remaining three were all *Candida*-negative [7].

In a recent review by Robledo-Sierra et al., the authors found only one other paper investigating the relationship between an OPMD (oral lichen planus (OLP)) and fungi. It was a pilot study on probiotic use in recurrent oral candidiasis [20].

Another review by Sami et al. deals with oral mycobiome in more detail [1]. After acknowledging the reduced richness and diversity of fungal species in OSCC, the authors unveiled the mycobiome’s implication in premalignant disorders, which is ambivalent. In fact, on one hand, *Malassezia*, *Schizophyllum* and *Emericella* have shown anti-carcinogenic potential; on the other hand, a few *Candida* species (*albicans*, *dubliniensis*, *tropicalis*, *pintolopesii* and *glabrata*) and *Saccharomyces cerevisiae* have been isolated from chronic hyperplastic candidiasis, a premalignant lesion with high rate of dysplastic transformation. *Candida* has also been associated with the severity of oral dysplasia. It was shown to produce either carcinogens (i.e., N-nitrosobenzylmethylamine and acetaldehyde) or proteinases able to degrade the basement membrane, particularly at an acidic pH, which is typical of cancer. The authors speculate on a synergistic effect of fungi and bacteria in tumour development, supported by the recent finding of a protective role of *Candida albicans* towards *Porphyromonas gingivalis* [24]. Lin et al. drew the same conclusions, suggesting a tight relationship between bacteria and *Candida* in the regulation of mucosal immunity, particularly T-reg and Th17 cells [2]. The authors also state that significantly higher abundances of *Candida* and *Aspergillus* were observed in patients with erosive OLP. Finally, they include *Candida albicans* in the pathogens involved in OSCC progression, together with the ubiquitous *Fusobacterium nucleatum* and *Porphyromonas gingivalis*.

Despite being a current topic of study in cancer development, particularly in the head and neck, viruses seem an outcast in the oral microbiome. None of the papers analyzing OLK and microbiome mentions the viral component; this finding agrees with the results of the review by Robledo-Sierra et al. [20]. Healy et al. highlight the lack of definitive association between Human Papillomavirus (HPV) and oral cancer. Still, they nonetheless recommend that thorough analysis of the virome in oral malignant and potentially premalignant tissues is carried out in the future because of the unknown HPV subtypes and possibly new viral pathogens [21]. As stated by Lin et al., “to date the roles of bacterial-fungal-virus interactions in OLP pathogenesis remain largely uncharacterized” [2]. Two recent meta-analyses explored the association of HPV with OLKs, and both groups encountered considerable heterogeneity between studies. Shang et al. found OLK to have a 2.5-fold increased association with HPV, particularly serotypes 16 and 18 [25]; de la Cour et al. discovered a 20.2% pooled prevalence of HPV in OLKs, with HPV16 being the most common genotype detected [26].

Finally, archaea are single-celled, obligate anaerobic microorganisms, able to live in extreme conditions, often detected in periodontal pockets. They are known to create a syntrophic environment with fermenting bacteria and favor their growth, but their role in OLKs has not been investigated.

### 2.4. The “Bad Fellas”: Tobacco, Alcohol, Betel and Chronic Mucosal Inflammation

Tobacco, alcohol and betel quid (or areca nut) are well-established synergistic risk factors for OLK and OSCC. Because of the mucosal damage caused by their action and the ensuing altered ecological niche, they act as confounders when investigating the oral microbiome. In particular, smoking has already been associated with the altered oral microbiome in a large-cohort American study [27].

Sami et al. reviewed the effects of tobacco (both smoking and smokeless) and alcohol on oral mucosa: smokers show a shift to a more pathogenic environment, a loss of commensal bacteria with a protective role, such as *Neisseria species*, and a reduced response to *Porphyromonas gingivalis*; alcohol consumers have an increased salivary acetaldehyde production and a decreased concentration of *Lactobacilli*, which can break down this carcinogenic compound [1]. Interestingly, the authors also state that drinkers have increased *Neisseria* genera, which produce acetaldehyde and may thus turn into commensal pathogens among this subgroup. Once more, the latter finding highlights that results are all but straightforward and caution is always needed when evaluating outcomes.

Possibly, for this reason, Ganly et al. included only non-smokers in their study and adjusted results for alcohol use, which was proven not to influence their findings [16].

Amer et al. found that smokers had reduced levels of *Neisseria* species, *Fusobacterium nucleatum* and *Leptotrichia*, and they stated that the effect of smoking on the oral microbiome warrants further investigation because non-smokers are more likely to undergo malignant transformation [7]. The same study showed that alcohol consumption was associated with enrichment in *Campylobacter* species. In the following paper, the same authors investigated the relationship between alcohol and *Rotha mucilaginosa*, which increased OLK. They showed that most of these bacteria could not metabolize acetaldehyde. Together with a decrease in acetaldehyde-dehydrogenase-producing *Streptococci*, they contribute to a higher exposure to this carcinogenic metabolite, possibly supporting the development of OLK and/or its malignant transformation in alcohol consumers [9].

Chewing of areca (betel) nut is practiced by 10 to 20% of the world population, being very common in the densely populated South and Southeastern Asia [13,14], and it is mainly related to the development of oral submucous fibrosis [4]. A couple of studies investigated the impact of this substance on the oral microbiome and oral premalignant lesions. Hernandez et al. studied 101 subjects exposed to betel and found that chewers had reduced richness and evenness of common oral bacteria, but this alteration was reversible after ceasing the exposition. They also discovered that chewers with oral lesions (leukoplakia and submucous fibrosis) had significantly elevated levels of *Oribacterium*, *Actinomyces* and *Streptococcus*, including *Streptococcus anginosus*. This somehow differs from previous findings of betel nut’s antibacterial properties, particularly against *Streptococci*. In the end, the authors wisely conclude that the influence in carcinogenesis of an altered oral microbiome in betel nut chewers is only speculative [13]. Zhong et al. reported modifications of oral microbiota by areca nut, yet with different effects depending on the subsite (i.e., tongue dorsum, buccal mucosa and gingiva).

For this reason, the authors question the results from Hernandez et al., who based their study on salivary samples. The Chinese authors also found decreased *Fusobacterium* and *Rothia* at both the buccal mucosa and gingiva in OSF compared to healthy areca chewers. Of note, this complex and confusing analysis did not include patients with leukoplakia and, most importantly, the different groups were not matched, and adjustment for smoking and alcohol was not carried out (i.e., healthy areca chewers, areca chewers with OSF and areca chewers with OSCC). Additionally, despite each stage of pathological progress bearing distinct alterations in the oral microbiota, no continuous changes were observed [14].

In a few reports, ahead of transformation into carcinoma, dysplasia of the oral cavity was related to alteration of the oral microbiota because of chronic mucosal inflammation. In turn, chronic mucosal inflammation relates to occlusal trauma: occlusal discrepancies and periodontitis are closely connected to the microbiological flora [12,28]. Oral microbiome, as well as oral chronic traumatisms, chronic periodontitis and poor oral hygiene, are reported to be involved in the carcinogenesis process in the oral cavity [29]. Of note, the inflammation in periodontal disease may be further worsened by nutrient deficiency, especially minerals and vitamins (A, B, C, D), which are necessary for teeth growth and mineralization and for balancing oxidative stress, respectively. Under these impaired conditions, a pathogenic microflora flourishes [30].

Moreover, an alteration of the oral microbiota is reported in the mucosal inflammatory responses in oral lichen planus, a chronic inflammatory autoimmune disease not defined as dysplasia but correlated to oral malignant transformation [31,32,33,34].

### 2.5. “What Are They Doing?”: Transcriptomics, Proteomics and Metabolomics

Up to date, all studies available on the oral microbiome and OLK and/or OSCC have focused on the identification and quantification of microorganisms present in a specific environment, thanks to the extraordinary capability provided by NGS [20]. This means that we now know “who is there”, but we still ignore “what they can do” and, more specifically, “what they are doing” [33]. Comprehensive analysis of the genomic transcripts (mRNA), proteins produced and end-product of bacterial metabolism would shed light on the function and fluctuating activity of the oral microbiome. Of course, such an analysis would prove highly complicated because of several factors, including RNA denaturation, variable transcriptional activation and the presence of human proteins and metabolites. Still, it may well explain the nature of pathobionts (i.e., commensal pathogens able to cause disease) and the dynamics of different bacterial clusters.

To unravel the relationship between cancer and precancerous lesions, studies on the oral microbiome should further evolve from taking static pictures of the ecologic landscape to videotaping the constant dynamic interaction of its protagonists.

We acknowledge that our work is limited by the small number of papers retrieved. Nonetheless, in our opinion, a narrative review may potentially overcome this limitation because it allows expansion of the analysis beyond the strict rules of a systematic review.

## 3. Conclusions

The studies available have proven an association between oral leukoplakia and alteration of the oral microbiota. However, the results are sometimes inconsistent and limited by sampling modality (i.e., swab vs. rinse vs. biopsy) and methodology, especially in terms of inclusion criteria (i.e., leukoplakia diagnosed clinically vs. histologically).

Bacterial richness and diversity tend to be decreased in OLK compared to healthy controls, with a reduction in the prevalent commensals, such as *Streptococci*, and an increase in anaerobes. Among the latter, *Fusobacterium nucleatum*, *Porphyromonas gingivalis* and *Prevotella intermedia* are recurrent findings, and they have already been linked to periodontal disease. Nonetheless, the disease is unlikely related to a single bacterium, and we should instead consider bacterial clusters with tangled interactions that are either constructive or destructive. These microbial community changes may also represent a marker for the transition from OLK to OSCC [8].

In the end, the “chicken or the egg” dilemma remains unsolved because the association does not mean causality: unhealthy hypoxic mucosa may be colonized by a bacterial flora different from normal conditions. Additionally, the “who-came-first” concept is probably simplistic because the relationship between the microbiome and precancerous lesions is complex. Diverse factors concur along the process of malignant transformation, as hypothesized by Healy et al. [21]. To shed light on this relationship, longitudinal studies are necessary: on one side, they should add investigation of microorganisms other than bacteria, especially fungi (e.g., *Candida* species), which have been overlooked so far; on the other side, they should include analysis of functional data, such as RNA transcripts, proteins and metabolites.

## Figures and Tables

**Figure 1 cancers-13-04439-f001:**
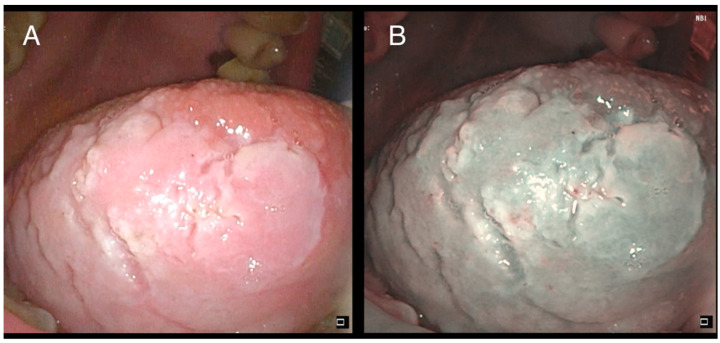
Low-grade dysplasia of the right tongue border: (**A**) white light; (**B**) narrow-band imaging (NBI).

**Table 1 cancers-13-04439-t001:** Original studies available on oral leukoplakia and the microbiome up to May 2021.

Author	Number of Samples	Number of Patients with OSCC	Number of Patients with Premalignant Lesions	Type of Premalignant Lesions	Definition of Premalignant Lesions (Histological/ Clinical)	Microbiological Sampling	Healthy Controls	Other Carcinogenic Factors	Main Microbiological Results
Hu et al. [15]	45	16	10	Leukoplakia	Clinical	Saliva collection	19	-	The genus *Streptococcus* was the most abundant in all three groups while *Neisseria* was the second most abundant Streptococcus and Abiotrophia the most abundant in the HC group. Haemophilus was much more abundant in the OLK group than in the OSCC, while Bacillus was the most abundant in the OSCC group.
Amer et al. [7]	68	-	36	Leukoplakia	Histological	Mucosal swab	23	Smoking, alcohol, oral hygiene, denture	The species most enriched in OLK include *Fusobacterium*, *Leptotrichia*, *Campylobacter,* and *Rothia* species.
Hernandez et al. [13]	122	-	10	Leukoplakia, erythroplakia, and submucous fibrosis	Clinical	Mucosal swab + saliva collection	-	Smoking, alcohol, chewing betel nut	*Streptococcus infantis* increased in current chewers compared to past/never chewers of betel nut. *Streptococcus anginosus* was increased in betel nut chewers with oral lesions compared to individuals with no lesions.
Lee et al. [8]	376	125	124	Dysplasia, hyperplasia, and hyperkeratosis	Clinical	Saliva collection	127	Smoking, alcohol	*Alistipes*, *Bacteroides*, *Blautia*, *Clostridium*, *Dorea*, *Escherichia*, *Faecalibacterium*, *Megamonas*, and *Phascolarctobacterium* displayed positive correlations with each other in the epithelial precursor lesion and cancer groups.
Decsi et al. [11]	7	-	7	Leukoplakia, lichen reticularis, lichen atrophicans	Histological	Tissue biopsy + mucosal swab	7	Smoking, alcohol	Increased *Fusobacterium nucleatum* and decreased *Streptococcus mitis* in patients with premalignant lesions.
Ganly et al. [16]	38	18	8	Leukoplakia	Histological	Saliva collection	12	-	OSCC patients showed enrichment in *Fusobacterium, Prevotella,* and *Alloprevotella* and depletion in *Streptococcus. Fusobacterium* and *Veillonella* were more abundant in patients with premalignant lesions than in the controls. An association of Capnocytophaga with OSCC recurrence was shown.
Hashimoto et al. [17]	16	6	6	Leukoplakia	Histological	Saliva collection	19	Smoking, alcohol	*Solobacterium* was increased in OSCC. A decrease in the abundance of the genus *Streptococcus* in patients with OSCC when compared with those with OLK was evaluated. *P. gingivalis* and *S. anginosus* increased in the OSCC and OLK groups.
Gopinath et al. [12]	74	31	20	Leukoplakia	Clinical	Mucosal swab + saliva collection	23	Smoking, alcohol, chewing betel nut, denture	OLK patients exhibited a decrease in *Firmicutes* and an increase in *Bacteroidetes.* The most variable genera between the OLK and OSCC groups were *Megaspheara, unclassified Enterobacteria, Prevotella, Porphyromonas, Granulicatella, and Salmonella.*
Shridhar et al. [18]	99	-	25	Leukoplakia	Clinical	Mucosal swab + saliva collection	74	Smoking, alcohol, chewing betel nut, denture	*P*. *gingivalis*, *F*. *nucleatum,* and *P*. *intermedia* were correlated among leukoplakia cases compared to the leukoplakia-free controls.

## Data Availability

The data presented in this study are available on request from the corresponding author.

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
