# Peer review of "Leukoplakia in the Oral Cavity and Oral Microbiota: A Comprehensive Review"

_cancers, 2021, doi:10.3390/cancers13174439_

Round 1

Reviewer 1 Report

The authors reviewed the current published literature on the impact of oral microbiota in oral cavity leukoplakia (OLK), aiming at clarifying its role in disease transformation. Unfortunately, authors reported that these studies present several limitations making an objective comparison difficult. The following points need to be revised and corrected. Reviewer expect the authors to take the comments seriously.

Major points

  1. The authors should add that bad dentition and the effects of dentition are associated with OLK. And so, add about the effects of vitamin A and B deficiency
  2. Authors must explain that candidiasis is also associated with diabetic white moss.
  3. Bacterial names and genus species must be italicized.
  4. Table 1 is extremely difficult to read. Correct the table so that it is easy for the reader to read, such as by swapping the vertical and horizontal directions.

Author Response

Thank you for your remarks, which were evaluated meticulously.

  1. The authors should add that bad dentition and the effects of dentition are associated with OLK. And so, add about the effects of vitamin A and B deficiency

We implemented the paragraph “The bad fellas” by highlighting the correlation between periodontal disease and nutrient deficiency.  

  1. Authors must explain that candidiasis is also associated with diabetic white moss.

We implemented the paragraph “The background” with a specific focus on candida lesions, particularly pseudomembranous candidiasis.

  1. Bacterial names and genus species must be italicized.

We modified the text accordingly.

  1. Table 1 is extremely difficult to read. Correct the table so that it is easy for the reader to read, such as by swapping the vertical and horizontal directions.

Table 1 was hard to read, indeed.  Thank you. We modified it accordingly.

Reviewer 2 Report

Well-written work, I think it's interesting how the authors named the chapters. In materials and method the authors declare a "systematic" review, but few lines below, they write about a "narrative" review. I think that the description of methods fits better with a narrative review.

The table is barely legible, probably is better to redesign it.

Author Response

Thank you for your remarks. Material and methods were unclear indeed, so we rewrote them completely. Table 1 was modified for a better visualization. 

Reviewer 3 Report

The authors performed a comprehensive review concerning "Leukoplakia in oral cavity and oral microbiota". Topic of the study is interesting in the field of oral medicine and oral pathology and could be of interest for the readership of the journal. I would like to recommend to extend the study design into systematic review and meta analysis including the guidelines for systematic reviews.

Author Response

Thank you for your remark. At the beginning of our work a systematic review was planned, but retrieval of articles was poor, thus we decided to change it into a narrative review. Nonetheless, we think that this type of review allows a better “expansion” of the topic and can provide valuable information as well. We rewrote material and methods in the text and further acknowledged the limitation of the study at the end of the discussion.

Reviewer 4 Report

The paper is a narrative review aimed to clarify the role, if any, of oral microbiota in oral leukoplakia.

The topic is interesting and update. The search was valuably done using three different databaseses. However the method for searching should me more detailed.

The authors finally select only ten  papers, almost dishomogeneous. Authors should clarify how many papers were selected and which was the decision tapering process which led to the final ten.

Author Response

Thank you for your remark. We rewrote the materials and methods to make our selection process clearer for the reader. 

Round 2

Reviewer 3 Report

The authors answered all questions and comments in detail.

Reviewer 4 Report

The Authors clarified methodology for searching. The apper is improved